# The Physical Developmental Characterization of Children with Nutritional Deficiencies and Attributed Specific Categories

**DOI:** 10.3390/nu17010086

**Published:** 2024-12-28

**Authors:** Jingjing Liu, Xinye Qi, Rizhen Wang, Junling Zhang, Shaoke Lu, Guangliang Xie, Yinghua Qin, Dongqing Ye, Qunhong Wu

**Affiliations:** 1School of Public Health, Anhui University of Science and Technology, Hefei 231131, China; jingjingliu2022@126.com (J.L.); zjunling062024@126.com (J.Z.); 17769555100@163.com (S.L.); 19505645985@163.com (G.X.); 2Department of Health Management, School of Public Health, Tianjin Medical University, Tianjin 300203, China; qixinye@tmu.edu.cn; 3Department of Global Health, School of Public Health, Peking University, Beijing 100191, China; wrz_99766@163.com; 4Department of Health Economy and Social Security, College of Humanities and Management, Guilin Medical University, Guangxi Zhuang Autonomous Region, Guilin 541199, China; qinyinghua250@126.com; 5Department of Social Medicine, School of Public Health, Harbin Medical University, Harbin 150081, China

**Keywords:** nutritional deficiencies, mean value of height for age, standardized values of weight for age, body mass index for age

## Abstract

Background: There are few studies examining the physical developmental phenotypes of nutritional deficiency diseases (NDDs) among Chinese children aged 1–7 years by anthropometrics and clarifying the specific NDD categories that caused growth faltering. Methods: A total of 3054 cases of NDDs in children aged 1–7 years were investigated. The age, height, and weight of children with NDDs were adjusted by using the skewness coefficient–median–coefficient of variation method, and the results were compared with the WHO standardized level. Comparisons of specific categories of NDDs were performed with respect to the age-specific height standardized values (HAZ), the age-specific weight standardized values (WAZ), and the age-specific body mass index standard values (BAZ). Result: The subtypes of NDDs among Chinese children were mainly characterized by single and co-morbid deficiencies. Calcium deficiency, vitamin A deficiency, vitamin D deficiency, and vitamin B deficiency were the main types, accounting for 11.33%, 9.26%, 8.70%, and 6.29% of the total confirmed cases, respectively; protein–energy malnutrition combined with vitamin C deficiency was the most common type of combined nutrient deficiency (5.76%). The HAZ (−0.0002), WAZ (−0.0210), and BAZ (−0.0018) of children aged 1–3 years with NDDs was lower than the WHO standard (0), as were the HAZ (−0.0003), WAZ (−0.0219), and BAZ (−0.0019) of children aged 3–7 years. The anthropometrics of children with NDDs aged 1–7 years showed that the HAZ and BAZ were slightly lower than the WHO average level, whereas the WAZ was significantly different from the WHO average. The co-morbidity of vitamin A deficiency and vitamin D deficiency, calcium deficiency and vitamin C deficiency, vitamin D deficiency and vitamin C deficiency, and iodine deficiency and vitamin C deficiency were associated with the WAZ. Interpretation: The specific categories of NDDs consist mainly of calcium deficiencies, vitamin A deficiencies, and vitamin D deficiencies. The main signs of growth retardation were low weight and height, which were driven by the specific single and co-morbid micronutrient deficiencies.

## 1. Introduction

Nutrition is fundamental to the growth and healthy development of children. In the context of a changing nutritional environment, unbalanced diets in children become a concerning problem; for example, the consumption of snacks, beverages, and prepared foods is associated with excessive energy intake, an unbalanced intake of nutrients, and micronutrient deficiencies that are often referred to as hidden hunger [1]. Child malnutrition, including overnutrition and undernutrition, has become a public health problem that requires urgent intervention, due to its magnitude and devastating consequences for children’s growth, development, and survival [2]. A top-level policy vision has been proposed both internationally and nationally to address the issue. In 2015, the UN Sustainable Development Goals enshrined the objective of “ending all forms of malnutrition”, challenging the world to think and act differently on malnutrition—to focus on all its faces and work to end it, for all people, by 2030.

However, progress on improving nutritional deficiency diseases (NDDs) in children remains slow. The concept of nutritional deficiency disease as defined by ESPEN is a nutritional state characterized by detectable changes in body tissues (body size, shape, and composition), decreased function, and adverse clinical outcomes caused by insufficient energy, protein, and other nutrients. The results of the latest assessment of the burden of NDDs in China by applying the global burden of disease data show that although the age-standardized overall incidence rate of NDDs in China has remained stable, there are large differences in the incidence rates of various age groups, and the incidence burden of NDDs in children is 4414/100,000, which is significantly higher than that of the other age groups (adolescents, young adults, the middle-aged, and the elderly) [3]. Currently, the severe trend of childhood NDDs poses a new challenge to addressing the double burden of malnutrition.

Previous studies focused more on the characterization of childhood obesity, micronutrient deficiency and their risk; however, the complex characterization of NDDs was underestimated. Suffering from obesity may be associated with NDDs. Studies in developing countries have shown that elevated systolic blood pressure and abdominal fat mass are associated with mild stunting in children at moderate to high levels, although there is no difference in the percentage of fat mass between mildly stunted and normal children [4]. In addition, fetal and early childhood NDDs result in a “thrifty phenotype”, such as small abdominal viscera and low muscle mass, with efficient fat storage [5]. Currently, the co-morbidity of obesity with micronutrient deficiencies has been demonstrated, such as the coexistence of obesity with anemia, zinc deficiency, and vitamin A deficiency [5]. In addition, although child stunting and wasting both indicate NDDs, they are also fundamentally distinct phenotypes with different timings and durations of the causal insults, specific risk factors, varied distributions across children, and different prognoses. Methodologically, anthropometrics is the most universally used method to assess children’s nutritional status. The phenotypes for children with NDDs can be determined through anthropometric indicators to improve their nutritional status, health status, and functional impairment.

In the past years, the growing literature has identified associations between small size; early patterns of postnatal growth; and adult conditions as diverse as diabetes, cardiovascular disease, and schizophrenia. More research is needed to provide evidence on the current growth and development patterns of children, especially in the context of the high disease burden of NDDs. In this study, we aimed to assess the physical developmental phenotypes of NDDs among Chinese children aged 1–7 years by anthropometrics, clarify specific NDD categories that caused growth faltering, and create the evidence base for comprehensive interventions for early physical development in children.

## 2. Methods

### 2.1. Data Collection

A combination of stratified and whole–cluster sampling was used as the sampling method for this study. First, provinces were sampled according to their geographic regions (east, central, and west) and economic income levels (High, 622018 billion; Medium, 2665.13 billion; and Low, 25,698.5 billion) [6].

We assigned a number to each of the 10 provinces/municipalities in eastern China, 6 provinces in central China, and 12 provinces/autonomous regions/municipalities in western China, and selected two provinces at random. These provinces were Guangdong Province and Zhejiang Province in the eastern region, Anhui Province and Jiangxi Province in the central region, and Gansu Province and Yunnan Province in the western region. Of the 6 provinces, we randomly selected two tertiary hospitals per province. A 1-month study was conducted from mid-November 2022 to mid-December 2022. All children aged 1–7 years diagnosed with NDDs in the target hospitals were enrolled in the study by cluster sampling, with children and caregivers who did not wish to participate excluded.

Based on the principle of voluntary participation, three trained nurses from 12 tertiary hospitals in 6 provinces recorded the basic information on nutritional deficiency diseases in children aged 1–7 years in the outpatient and inpatient departments, with the main information including the type of nutritional deficiency disease, and asked the caregivers to report the length and weight of the child at birth as well as the child’s current age, height, and weight.

Children diagnosed with NDDs were included in the survey directly from either the outpatient or inpatient department by asking the children if they suffered from the subtypes of protein–energy malnutrition, iron deficiency, etc., which had been definitively diagnosed by the doctors. Actual medical testing was not carried out again. This basic information was part of a complete questionnaire on multidimensional parenting, cultural, and ecological factors influencing childhood nutritional deficiency disease. A total of 3054 cases of NDDs in children were collected in this study.

### 2.2. Statistical Analysis

According to the definition of WHO child growth standards, the weight-for-age (WAZ) was used to determine the immediate and long-term nutritional status of children. A WAZ value 3 standard deviations below the population median (−3 SD) was considered a severe nutritional deficiency disease, while children with HAZ values that were 2 standard deviations below the median (−2 SD) were categorized as moderately malnourished. Height-for-age norms (HAZ), which are an indicator of linear growth retardation and cumulative growth deficit, were used to examine chronic malnutrition in children, with height-for-age norms 3 standard deviations below the median defined as severe stunting and children 2 standard deviations below the median (−2 SD) categorized as moderately stunted. Similarly, children with Z-scores for height-for-age below the median of the reference population minus 2 standard deviations were considered to be short for their age or stunted and chronically malnourished. When the Z-score was less than −3 standard deviations, the child was categorized as severely stunted [7]. Stunting reflects a chronic failure to receive adequate nutrition. The standardized value of the weight-for-age group took into account both acute and chronic malnutrition; values that were 2 standard deviations but not more than 3 standard deviations below the median growth curve (−3 ≤ SD < −2) were classified as low birth weight, and the weight-for-age group more than 3 standard deviations below the growth curve (−3 SD) was classified as severe low birth weight. National and international measurements of micronutrient-related NDDs often rely on a variety of physiologic medical assays and laboratory tests for their determination. For example, hemoglobin iron levels test for anemia, and the levels of zinc, iodine, and various types of micronutrients such as vitamin A and vitamin B test for healthy levels of body mass. NDDs are determined based on whether the micronutrient test values are in the normal range, and any values below the normal range are defined as micronutrient-related malnutrition.

To examine the growth status of children with NDDs, this study was conducted in accordance with WHO’s standards, and standardized values were created for the weight, height, and body mass index of children with NDDs based on relevant work practices, scholars’ suggestions [8], and the requirement of inter-study comparability [9]. In accordance with the technical information provided by WHO, the standardized values were created using the coefficient of skewness–median–coefficient of variation (LMS) method. The conversion formula is as follows:$$Z=\frac{(X/M)L-1}{LS},L\neq 0$$
where Z is the standardized value, X is the sample measurement, M is the mean value of the growth standard, S is the adjusted standard deviation coefficient, and L is the Box–Cox transformation coefficient. The parameters of L, M, and S were provided by the official reports of the respective growth standardization departments. In addition, in order to adapt to the parametric standards, the age of the samples in the study was reported as “age in months” when creating the standardized values.

The analysis of children’s growth relies on the measurements for two variables: height and weight [8]. Using the above formula and the parameters of the growth curves provided by WHO, the indicators used in this study for converting the sample data to standardized values are as follows:

Age-specific weight standardized values (WAZ), which were used to determine the immediate and long-term nutritional status of children;

Age-specific height standardized values (HAZ), which were used to examine chronic NDDs in children;

Age-specific body-mass-index standard values (BAZ) [body mass index (BMI) = weight (kg)/height squared (m^2^)], which were used to determine whether children were overweight or obese.

The following terms were used to describe certain values of the standardized growth variables:

a. Growth retardation: the HAZ was more than 2 standard deviations below than the median WHO growth curve (HAZ < −2);

b. Low body weight: the WAZ was more than 2 standard deviations but not more than 3 standard deviations below the median growth curve (−3 ≤ WAZ < −2); Severe low body weight: the WAZ was more than 3 standard deviations below the median (WAZ < −3);

c. Wasting: the BAZ was more than 2 standard deviations but not more than 3 standard deviations below the median growth curve (−3 ≤ BAZ < −2); Severe wasting: the BAZ was less than 3 standard deviations below the median growth curve (BAZ < −3);

d. Overweight: the BAZ was greater than 2 standard deviations below the median growth curve but not more than 3 standard deviations below it (2 < BAZ ≤ 3); Obesity: the BAZ was greater than 3 standard deviations below the median growth curve (BAZ > 3).

The subtypes of the NDDs were reported using general statistical descriptions of frequencies and percentages. The transformation of the LMS–standardized values for growth were performed using SAS 9.4. Comparisons of specific categories of NDDs with respect to the HAZ, WAZ, and BAZ were done using an independent samples *t*-test. The statistical significance level of the test was expressed as α = 0.05. Statistical analyses were performed using SPSS, version 22.

## 3. Results

### 3.1. Distribution of Types in Children with NDDs

Among the 3054 values in the sample, the subtypes of NDDs were mainly characterized by single and co-morbid deficiencies. Of all subtypes of NDDs in children, calcium deficiency, vitamin A deficiency, vitamin D deficiency, and vitamin B deficiency were the main types, accounting for 10.4%, 9.2%, 8.7%, and 6.4% of the total number of confirmed cases, respectively; protein–energy malnutrition combined with vitamin C deficiency was the most common type of combined nutrient deficiency (5.76%), as shown in Figure 1.

### 3.2. Characterization of Growth in Children with NDDs

From the definition of growth among children, it can be seen that if the standardized value of an indicator was greater than 0, it means that the measurement of that indicator was higher than the average in the WHO growth charts, and conversely, if the standardized value was less than 0, the measurement of that indicator was lower than the WHO average.

Figure 2 showed that the mean value of the HAZ for children aged 1–3 years (including 3-year-olds) was a standardized score of −0.0002, and the WAZ and BAZ standardized scores were −0.0210 and −0.0018, respectively, indicating that the height development and the weight development among children with NDDs were all lower than the WHO average. The HAZ, WAZ, and BAZ of children aged 3–7 years were −0.0003, −0.0219, and −0.0019 respectively, indicating that children aged 3–7 years had the same external physical developmental phenotype of low height and low weight as children aged 1–3 years among children with NDDs.

### 3.3. Comparison of Growth Characterization in Specific NDDs

Figure 3 described the specific categories of NDDs with respect to their HAZ, WAZ and BAZ values. The co-morbidity of vitamin A and vitamin D deficiency, vitamin D and vitamin B deficiency, calcium and vitamin C deficiency, vitamin D and vitamin C deficiency, and iodine and vitamin C deficiency were associated with WAZ scores that were significantly different from the WHO average (*p* < 0.05). In addition, the WAZ score was also affected by the single disease of iodine deficiency (*p* < 0.05). As for the HAZ among children with NDDs, the co-morbidity of calcium and vitamin D deficiency, vitamin D and vitamin B deficiency, vitamin A and iodine deficiency, protein–energy malnutrition and vitamin D deficiency, and iron and vitamin B deficiency, as well as a single calcium deficiency or an iodine deficiency, were all associated with differing scores.

## 4. Discussion

We made a preliminary assessment on the physical developmental phenotypes of NDDs among Chinese children aged 1–7 years by anthropometrics and clarified specific NDD categories that caused growth faltering. The specific categories of NDD among Chinese children were mainly calcium deficiencies, vitamin A deficiencies, and vitamin D deficiencies. The anthropometrics of children with NDDs aged 1–7 years showed that the HAZ and BAZ were slightly lower than the WHO average level, whereas the WAZ was significantly different from the WHO average. The specific category that contributed to the growth faltering of children with NDDs in China manifested as combined micronutrient deficiencies.

According to the findings of our study, the main types of NDDs in Chinese children were calcium deficiency, vitamin A deficiency, and vitamin D deficiency, accounting for 11.33%, 9.26%, and 8.70% of the total number of confirmed cases, respectively. The prevalence of these three vitamin deficiencies in Chinese children has also been confirmed in other studies [10,11]. The latest epidemiological survey data showed that the overall nutritional level of vitamins A and D in children was not up to the standard, with the overall prevalence of vitamin A deficiency in children aged 3–5 years at 29.3%, and the prevalence of vitamin D deficiency and insufficiency at 51.9% [12]. Calcium, vitamin A, and vitamin D deficiencies in China are associated with dietary supplements and unbalanced diets, and the International Osteoporosis Foundation “Interactive Map of Global Dietary Calcium Intake” shows that China’s per capita daily dietary calcium intake was the sixth lowest in the world.

China has made a lot of efforts in the prevention of and intervention in calcium, vitamin A, and vitamin D deficiencies. The “Dietary Guidelines for Chinese Residents” (2022) recommend that children’s calcium intake should be 300–500 g. The best way to supplement calcium is through the daily diet, such as with milk, soya bean products, cheese, leafy greens, fish, shrimp and seaweed, etc. However, most parents are too busy to give their children milk and balanced diets on a daily basis. As most parents are too busy to give their children daily milk supplements and a balanced diet on time, calcium supplements were the most common and important way of calcium supplementation nowadays. However, a single calcium supplement does not have a significant effect and needs to be combined with vitamin D supplements. The “Dietary Guidelines for the Chinese Population” (2022) combines poultry, meat, aquatic products, and eggs into a single animal food, emphasising the need for both calcium and vitamin D supplementation. Strategies for the prevention and treatment of vitamin A deficiency diseases usually include increasing vitamin A intake through dietary diversity, vitamin A fortified food placement, and the use of vitamin A supplements, the latter of which has become the public health strategy for managing vitamin A deficiencies in China; however, the expected effects of vitamin A supplementation might vary in different regions and populations, which might be due to differences in the extent of vitamin A deficiency or the availability of other nutrients.

Other than parents’ inability to provide children with a balanced diet in the context of China’s rapid economic and social development, the varied dietary habits in different parts of China, as well as varying sunlight and UVR exposure may be partly responsible for the differences in micronutrient deficiency. In addition, due to the unbalanced development of urban and rural areas, the development level of Chinese children in urban and rural areas showed a trend of unbalanced development [13,14]. In the “China Child Development Report” (2022), it was pointed out that the average growth level of urban children has reached or even exceeded the WHO standard, and is close to the average level of children of the same age in Western developed countries. At the same time, there is still a gap between the growth standards of rural children and those of urban children, and they still have high physical developmental potential [14].

The height index of Chinese children with NDDs was slightly lower than the WHO normal standard. The height attained was the result of the interaction between genetic endowment and macro- and micronutrient availability during the growth period. Nutrition plays a key role in controlling linear growth through a process of cell proliferation, the addition of new cells to the growth plate of the bone, and hypertrophy. Protein–energy malnutrition caused by inadequate intakes of dietary energy and protein are known causes of growth retardation [15,16]. More recently, however, the role of specific micronutrient deficiencies in the aetiology of growth retardation has received increasing attention [17,18]. According to the study, the co-morbidity of calcium deficiency and vitamin D deficiency, vitamin D deficiency and vitamin B deficiency, vitamin A deficiency and iodine deficiency, and iron deficiency and vitamin B deficiency, as well as the single diseases of calcium deficiency and iodine deficiency, were all associated with less height growth among children. Evidence from human investigations indicates that vitamin D and calcium deficiencies also affect bone development, as manifested through the condition known as rickets [19]. Vitamin A was first identified as the growth-promoting factor “A.” Studies in the 1920s–1930s demonstrated arrested growth, especially of weight in rats, after acute vitamin A depletion [20,21]. Several micronutrients, including iodine, iron, and vitamin A, were also associated with immune function and the risk of morbidity, which in turn affect growth [22].

Notably, the anthropometric weight among children with NDDs aged 1–7 years was significantly different from the WHO average, which was mainly caused by the co-morbidity of vitamin A and vitamin D deficiency, vitamin D and vitamin B deficiency, calcium and vitamin C deficiency, vitamin D and vitamin C deficiency, and iodine and vitamin C deficiency. Calcium, vitamin D, vitamin B, iodine, iron, and vitamin A are associated with immune function, with deficits increasing exposure to infectious diseases and the risk of morbidity [22]. Because of inadequate dietary intakes and recurrent infectious diseases, children display a reduced appetite, increased metabolic requirements, and increased nutrient loss, and thus children falter in their weight. Previous efforts to prevent weight loss focused on diseases rather than on improved child feeding practices as such. However, Becker, Black, and Brown point out improving dietary intakes is more effective than disease prevention efforts in reducing NDDs [23]. The optimal interventions should aim at feeding healthy children optimal diets, which includes paying attention to dietary quality.

Improving early physical development in children who suffer from NDDs and promoting healthy human capital formation require many efforts. Feasible suggestions are as follows: (1) the evaluation of the effectiveness of national micronutrient deficiency fortification programs to reduce NDDs, especially for calcium, vitamin D, vitamin B, iodine, iron, and vitamin A; (2) the promotion of the optimal feeding of infants and young children through health education, professional support, lay support, and media campaigns; (3) the promotion of optimal dietary quality in children through health education, professional support, lay support, and media campaigns. Nutritional deficiency diseases increase the likelihood that a child will be sick and, when sick, will become seriously ill. Thus, resources must be allocated to health care services to deal with the increased frequency and severity of illness caused by undernutrition and micronutrient deficiencies.

This study increased our understanding of the physical developmental phenotypes of NDDs among Chinese children aged 1–7 years and clarified specific NDD categories that caused growth faltering. The findings could prove useful to assessing progress in the fight against all forms of malnutrition and serve as an important resource for the planning, targeting, monitoring, and evaluating of health promotion programs. This study also had some limitations. The recall bias and selection bias in the cross-sectional population study cannot be eliminated completely. A crucial work in the future would be the inclusion of large prospective cohort studies to substantially reduce recall bias and selection bias. The children were from different geographical locations of China, which differ in terms of economic development, dietary habits, and culture of upbringing. This study cannot consider the effects of economic development, dietary habits, and culture of upbringing; however, other studies have already reported these effects [24].

## 5. Conclusions

In China, calcium, vitamin A and vitamin D were commonly deficient in children aged 1–7 years who suffered from NDDs, and among these children, the main signs of growth retardation were low weight and height. The several specific micronutrients with the strongest relationship to children growth who suffered from NDDs included the co-morbidity of calcium deficiency, vitamin A deficiency, vitamin D deficiency, vitamin B deficiency and vitamin C deficiency, as well as single deficiencies in calcium and iodine.

## Figures and Tables

**Figure 1 nutrients-17-00086-f001:**
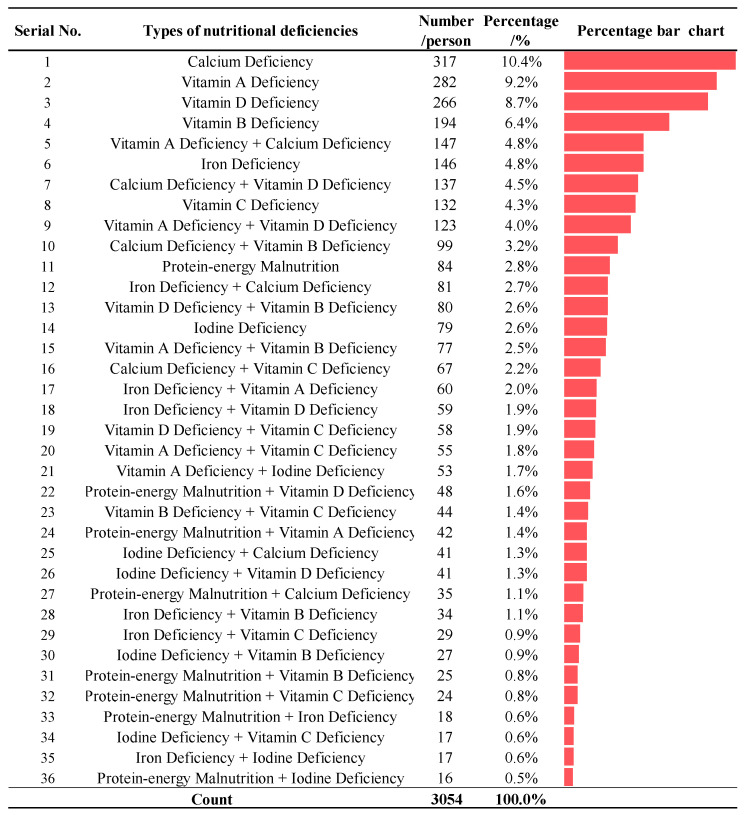
Distribution of diagnosed types of NDDs in children. **Note:** The coloured bars indicated the percentage of each type of nutritional deficiency.

**Figure 2 nutrients-17-00086-f002:**
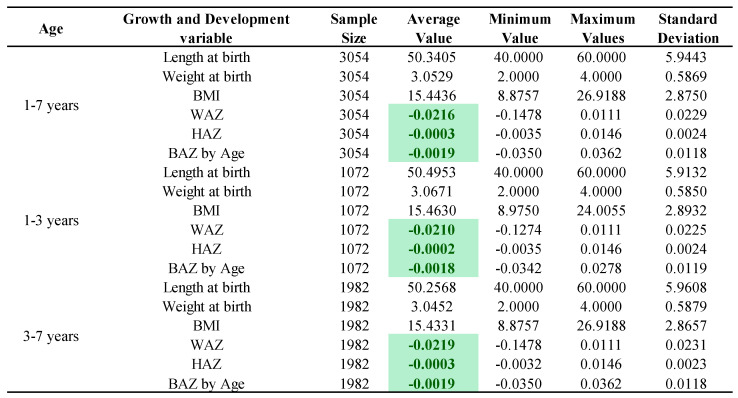
Growth characterization of children with NDDs by age. **Note:** The 1–3 years group includes 3-year-olds and the 3–7 years group does not include 3-year-olds; the green markings indicate that the growth among children with NDDs was lower than the WHO average.

**Figure 3 nutrients-17-00086-f003:**
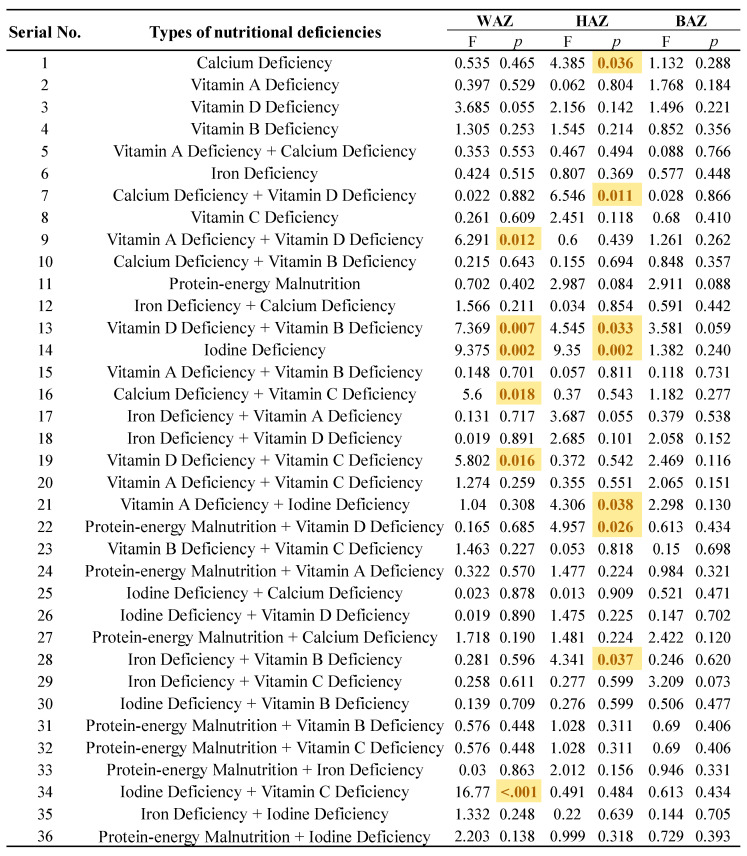
Comparison of growth characterization in specific NDDs. **Note:** The yellow markings indicate the specific categories where growth among children with NDDs was affected (*p* < 0.05).

## Data Availability

The original contributions presented in this study are included in the article. Further inquiries can be directed to the corresponding authors.

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
