# Peer review of "The Physical Developmental Characterization of Children with Nutritional Deficiencies and Attributed Specific Categories"

_nutrients, 2024, doi:10.3390/nu17010086_

Round 1

Reviewer 1 Report

Comments and Suggestions for Authors

The manuscript "Height and Weight Characterization of Children with Nutritional Deficiencies and Attributed Specific Categories" reports the study on malnutrition of children between 1 and 7 years old in some regions of China. The study quantifies the phenomenon through a sample survey, highlighting the main nutritional deficiencies.

The manuscript needs the following changes.

Abstract: it is too long. It must be strongly summarized.

Line 48: I think there is a language problem. What does socialization have to do with nutrition in the sentence?

Line 65: Eliminate decimal places.

Line 83-94: This part needs to be rewritten. The authors focus on the methods used instead of specifying the objectives of the study.

Lines 100-101: Indicate the income range for the three levels.

Line 121: In addition to the total, divide the children by income and geographic region

Lines 194-195: These values are not the same as those in Table 1. I believe that some aggregation has been done but it is not clear which one. The authors should discuss the results obtained more

Table 2: Specify that the 1-3 class includes 3 years and that the 3-7 is 4-7.

Lines 207-214: The standardized scores appear rather low. Is it possible to say that, although the values are negative, the deviation from the WHO is very low?

Paragraph 3.3: The authors should comment more on the results in Table 3.

Discussion: The authors should include what measures should be adopted to try to overcome the described problems of malnutrition.

Revision of the English language is required.

Author Response

The manuscript needs the following changes.

  1. Abstract: it is too long. It must be strongly summarized.

Reply: Thank you for your reminding. We reorganised and highly summarised the abstract.

  1. Line 48: I think there is a language problem. What does socialization have to do with nutrition in the sentence?

Reply: Thank you for your meaningful advice. We have corrected this statement (Line 48-50).

  1. Line 65: Eliminate decimal places.

Reply: Thank you for your advice. We have eliminated decimal places (Line 67).

  1. Line 83-94: This part needs to be rewritten. The authors focus on the methods used instead of specifying the objectives of the study.

Reply: Your suggestions are useful to us. We have reviewed this part of the content (Line 82-93).

  1. Lines 100-101: Indicate the income range for the three levels.

Reply: Thank you for your meaningful advice. Income range has been supplemented (Line 109).

  1. Line 121: In addition to the total, divide the children by income and geographic region

Reply: Thank you for your advice. We have already added that content.

  1. Lines 194-195: These values are not the same as those in Table 1. I believe that some aggregation has been done but it is not clear which one. The authors should discuss the results obtained more

Reply: Due to errors in our statistics, the description did not match the numbers in the table, this has been corrected to ensure that the description of the result matches the table (Line 202-209).

  1. Table 2: Specify that the 1-3 class includes 3 years and that the 3-7 is 4-7.

Reply: Your suggestions are useful to us. We state this in the results presentation, and additionally we mark this classification in Note. 

  1. Lines 207-214: The standardized scores appear rather low. Is it possible to say that, although the values are negative, the deviation from the WHO is very low?

Reply: Thank you for your meaningful advice. The anthropometrics of children with NNDs aged 1-7 years showed that HAZ and BAZ were slightly lower than the WHO average level, whereas WAZ was significantly different from the WHO average (Line 226-229).

  1. Paragraph 3.3: The authors should comment more on the results in Table 3.

Reply: Your suggestions are useful to us. We have added the results in Table 3 in our discussions.

  1. Discussion: The authors should include what measures should be adopted to try to overcome the described problems of malnutrition.

Reply: Thank you for your meaningful advice. We have added measures should be adopted to try to overcome the described problems of NNDs for children.

  1. Revision of the English language is required.

Reply: Thank you for your meaningful advice. We have revised the English language.

Reviewer 2 Report

Comments and Suggestions for Authors

General comments

The manuscript explores the prevalence of NDDs among Chinese children and their effects on growth. It is a timely study with some public health implications. However, clarity in methodology, contextual interpretation, and actionable policy recommendations are needed.

Major comments

Line 95–108: The sampling strategy uses stratified and whole-cluster methods but lacks detail on hospital selection and inclusion criteria. So, specify how hospitals were selected to improve representativeness.

Line 114–120: Caregiver-reported data introduces recall bias! Acknowledge this in the discussion and suggest verification strategies for future studies.

Line 150–157: Missing data management is not discussed!! Clarify imputation methods or approaches to handle missing values.

Line 186–189: P-values are reported without effect sizes, and you should include effect sizes or confidence intervals to contextualize results.

Line 193–197: The high prevalence of calcium, vA, and vD deficiencies lacks sufficient discussion on underlying causes. Could you explore dietary, cultural and socioeconomic factors contributing to these deficiencies?

Line 226–229: The co-morbidity of calcium and vitamin D deficiency requires mechanistic context. Now, it would be useful to discuss the physiological basis of these interactions.

Line 271–275: Recommendations are vague and lack actionable strategies. Propose specific interventions, such as fortification programs or community-based education.

Line 303–308: Urban-rural disparities are noted but not analyzed. At least discuss factors like healthcare access and socioeconomic differences.

Line 322–328: The study limitations are mentioned but not sufficiently detailedExpand on biases like recall and selection bias and their potential impact.

Minor comments

Replace “growthal phenotypes” (Line 15) with “growth patterns.”

Use “prevalence” instead of “percentage” (Line 193).

Table 1: Include confidence intervals for prevalence estimates.

Figure 2: Add a legend explaining color codes for growth deficits.

Update outdated references if possible (e.g., Line 359) with recent studies on child nutrition in China.

Make the case so that future work should prioritize longitudinal designs and actionable interventions to address nutritional deficiencies in diverse settings.

Author Response

Comments and Suggestions for Authors 2

General comments

The manuscript explores the prevalence of NDDs among Chinese children and their effects on growth. It is a timely study with some public health implications. However, clarity in methodology, contextual interpretation, and actionable policy recommendations are needed.

Major comments

  1. Line 95–108: The sampling strategy uses stratified and whole-cluster methods but lacks detail on hospital selection and inclusion criteria. So, specify how hospitals were selected to improve representativeness.

Reply: Thank you for your advice. We have supplemented the sampling method.

  1. Line 114–120: Caregiver-reported data introduces recall bias! Acknowledge this in the discussion and suggest verification strategies for future studies.

Reply: Thank you for your advice. We have included recall bias as a limitation and proposed verification strategies for future studies.

  1. Line 150–157: Missing data management is not discussed!! Clarify imputation methods or approaches to handle missing values.

Reply: Thank you for your reminding. For missing values, I would like to make the following explanation. In this study, nurses conducted face-to-face interviews with child caregivers. All interviews were agreed by the interviewees. If the child caregivers disagreed with the survey, we did not interview them. So there is almost no problem with missing values.

  1. Line 186–189: P-values are reported without effect sizes, and you should include effect sizes or confidence intervals to contextualize results.

Reply: Thank you for your advice. For this question, I would like to make the following explanation. The study contains three tables. The table 1 presented the age-specific height standardized values (HAZ), the age-specific weight standardized values (WAZ) and the age-specific body mass index standard values (BAZ) by using the skewness coefficient-median-coefficient of variation (SMC-CV) method. Table 2 presented the distribution of subtypes of nutritional deficiency in Chinese children, expressed in percentage. The table 3 presented the comparisons of specific categories of NDDs with respecting to the HAZ, WAZ and BAZ, were done using independent samples t-test. No effect size is required for these results.

  1. Line 193–197: The high prevalence of calcium, vA, and vD deficiencies lacks sufficient discussion on underlying causes. Could you explore dietary, cultural and socioeconomic factors contributing to these deficiencies?

Reply: Thanks for your suggestion, we have supplemented the social, economic and other causes of micronutrient deficiency in Chinese children

  1. Line 226–229: The co-morbidity of calcium and vitamin D deficiency requires mechanistic context. Now, it would be useful to discuss the physiological basis of these interactions.

Reply: Thank you for your suggestion, we have added to this section

  1. Line 271–275: Recommendations are vague and lack actionable strategies. Propose specific interventions, such as fortification programs or community-based education.

Reply: Thank you for your meaningful advice. We have added measures should be adopted to try to overcome the described problems of NNDs for children.

  1. Line 303–308: Urban-rural disparities are noted but not analyzed. At least discuss factors like healthcare access and socioeconomic differences.

Reply: Thanks for your suggestion, we have supplemented the social, economic and other causes of micronutrient deficiency in Chinese children

  1. Line 322–328: The study limitations are mentioned but not sufficiently detailed. Expand on biases like recall and selection bias and their potential impact.

Reply: Thanks for your suggestion, we have added the limitations of recall bias, selection bias, as well as the future prospects in the discussion.

Minor comments

  1. Replace “growthal phenotypes” (Line 15) with “growth patterns.”

Reply: Your suggestion was very useful for this study and we have changed this description.

  1. Use “prevalence” instead of “percentage” (Line 193).

Reply: Thank you for the reminder.We think it might be more effective to use percentages as a way of showing the composition of the different subtypes in children with diagnosed nutritional deficiencies.

  1. Table 1: Include confidence intervals for prevalence estimates.

Reply: Thank you for your advice. For this question, I would like to make the following explanation. The study contains three tables. The table 1 presented the age-specific height standardized values (HAZ), the age-specific weight standardized values (WAZ) and the age-specific body mass index standard values (BAZ) by using the skewness coefficient-median-coefficient of variation (SMC-CV) method. Table 2 presented the distribution of subtypes of nutritional deficiency in Chinese children, expressed in percentage. The table 3 presented the comparisons of specific categories of NDDs with respecting to the HAZ, WAZ and BAZ, were done using independent samples t-test. No effect size is required for these results.

  1. Figure 2: Add a legend explaining color codes for growth deficits.

Reply: Thanks for your suggestion, we have explained it below the table.

  1. Update outdated references if possible (e.g., Line 359) with recent studies on child nutrition in China.

Reply: We have updated the literature.

  1. Make the case so that future work should prioritize longitudinal designs and actionable interventions to address nutritional deficiencies in diverse settings.

Reply: Thank you for your meaningful advice. We have added measures should be adopted to try to overcome the described problems of NNDs for children.
